# Patient self-reported pain and nausea via smartphone following day care surgery, first year results: An observational cohort study

Bram Thiel[1]*, Jamey Blaauboer[2], Chiem Seesing[2], Jamshid Radmanesh[3], Seppe Koopman[4], Cor Kalkman[5], Marc Godfried[1]

1 Department of anaesthesia, OLVG Hospital, Amsterdam, the Netherlands, 2 Faculty of medicine, University of Amsterdam, Amsterdam, the Netherlands, 3 Department of information technology, OLVG Hospital, Amsterdam, the Netherlands, 4 Department of anaesthesia, Maasstad Hospital, Rotterdam, the Netherlands, 5 Department of anaesthesia, University Medical Centre Utrecht, Utrecht, the Netherlands

* b.thiel@olvg.nl

## Abstract

Contact with the hospital is usually limited for patients after day care surgery. Dedicated smartphone applications can improve communication and possibly enhance outcomes. The objective of this retrospective study was to evaluate patients' self-reported pain and nausea and assess the success of routine implementation of a smartphone application for outcome reporting. During preoperative assessment, patients were instructed to download and activate the smartphone application to report pain, nausea and to be in contact with the hospital after discharge. Main outcome was the number of patients actively using the smartphone application and the incidence of pain and nausea on postoperative day 1 to 7. In total, 4952 patients were included in the study. A total of 592 (12%) participants downloaded the application, of whom 351 (7%) were active users. A total of 4360 (88%) participants refrained from downloading the application. 56% (2,769) were female, the median age was 46 (18–92), and 4286 (87%) were classified as 1 or 2 American Society of Anesthesiologists Physical Status (ASA). Postoperative pain was experienced by 174 (76%) of 229 active users on postoperative day (POD) 1 and decreased to 44 (44%) of 100 active users on POD7. Postoperative nausea was experienced by 63 (28%) of 229 active users on POD1 and decreased to 12 (12%) of 100 active users on POD7. Female sex (p .000), socioeconomic status (p .001), and surgical severity (p .001) showed statistically significant differences between active users, non-active users, and non-downloaders. Most patients active with the application experienced pain and nausea on the first and second day after discharge. Only a minority of the patients used the application. Those who used it were satisfied with the possibilities offered to them. Future research should focus on increasing the uptake and effect of this application on the quality of recovery.

## Author summary

In the past, surgical recovery mainly occurred in hospitals. However, advancements in minimally invasive surgical techniques and anaesthesia have enabled us, to allow patients

**Data Availability Statement:** Data cannot be shared publicly because of the Netherlands general data protection regulation (GDPR). The data will be

stored for 15 years on a secured server at OLVG Hospital. For interested researchers it is possible to access the data by submitting an application to the local research advisory committee of OLVG Hospital, referencing the study with number WO 20.239 at acwo@olvg.nl.

**Funding:** The SIDN fund, a public benefit organization for Dutch Internet domain registration (URL: www.SIDNfonds.nl), partially funded the development of the smartphone application. The funders had no role in the study design, data collection and analysis, decision to publish, or manuscript preparation.

**Competing interests:** The authors have declared that no competing interests exist.

to recover at home after day care surgery. It is common practice that we provide patients with verbal and written instructions for managing their pain and nausea after discharge. Nevertheless, we have noticed that patients often face challenges when attempting to contact the hospital in case of severe pain or nausea, and as healthcare professionals, we often lack insight into their recovery at home. To address these issues and improve patient care, we implemented a smartphone application called the OLVG Pain app specifically for our day care surgical patients. The application empowers our patients to regularly report their pain and nausea scores and request consultations or adjustments to their medication as needed. Among the 4952 patients in our study, only a mere 7% actively used the app. We have found that postoperative pain and nausea were predominantly reported by the active users on postoperative day 1 and 2. The app's low uptake and usage suggest that there may be potential barriers related to the digital divide, particularly among our patient population. Further research is needed to explore these barriers, improve adoption rates, and assess the app's impact on postoperative recovery.

## Introduction

Previously, surgical recovery was typically performed in hospitals. Today, with the adoption of minimally invasive surgical procedures and modern anaesthesia techniques, patients can recover at home after surgery in day care. This shift has resulted in an increase in the number of day care procedures performed, as they are cost-effective and allow for earlier discharge [1]. In practice, discharge from day care surgery is possible as soon as vital signs are stabilized and the patient feels comfortable. The patient then receives verbal and written instructions on how to act in case of pain and nausea [2]. Moreover, the patient is discharged with a prescription for analgesics and antiemetic's, if deemed necessary. The importance of effective postoperative pain management has been demonstrated by a recent randomized controlled trial showing an association between a high pain level and poor or intermediate quality of recovery [3].

However, patients often encounter difficulties when attempting to contact the hospital by phone in cases of severe pain or nausea. In turn, healthcare professionals involved often have little insight into the recovery trajectory of their patients at home. Remote monitoring with a direct feedback loop between patients and healthcare professionals to tailor pain and nausea management could overcome these problems and improve clinical patient outcomes [4].

Since February 2020, we have provided day care surgical patients in our hospital with a smartphone application, the OLVG pain app, which allows them to report their pain and nausea scores regularly. Furthermore, they could use the app to request consultation with the hospital for advice or adjustment of their medication. This dedicated application was developed in collaboration with patients and evaluated in a proof-of-concept study of 50 hospitalized patients and 12 hospital stakeholders, such as anaesthetists and software engineers. The results showed that the smartphone application was user-friendly and had high satisfaction among patients and stakeholders, with outcomes comparable to pain assessments by nurses [5].

Despite the potential benefits of eHealth tools and mobile apps in patient care, their use is largely driven by optimistic rather than evidence-based assumptions [6]. The present study aimed to contribute to the evidence of mobile health by evaluating self-reported postoperative pain and nausea scores of patients using the app one year after its implementation in day care. In addition, we evaluated the uptake and actual use of the app by the patients to assess whether the routine provision of such a tool is feasible.

## Methods

### Study design and setting

From February 2nd 2020 to March 29th 2021, a retrospective observational cohort study was conducted in OLVG Hospital, a large teaching hospital with two locations in Amsterdam, the Netherlands. Annually, over 8.500 day care surgical interventions are performed in OLVG.

### Participants and perioperative anaesthesia practice

The following inclusion criteria were required: age >18 years and scheduled for day care surgery. The exclusion criterion was unplanned stay in the hospital after day care surgery.

During the preoperative assessment, information about the type of anaesthesia, medication, and the need for preoperative fasting was explained. The patient was also informed about the OLVG Pain app and received instructions for its use. A summary with the instructions was sent by e-mail or letter. Between January 2021 and March 2021, a researcher (CS) was available at the day care ward for assistance with downloading the app on the patient's smartphone and connecting it to their medical records.

After downloading the app, the patients entered their surnames, dates of birth, and gender. They were asked to consent to using their anonymized data for research purposes and to allow the app to send reminder notifications. Administrative staff verified the patient's identity and connected the app to their medical records in EPIC (1979–2022 Epic Systems Corporation, Wisconsin, United States). Hands-on instructions were provided by one of the researchers if required.

Anaesthesia, postoperative pain (POP) and postoperative nausea and vomiting (PONV) were managed according to the standards of the Dutch Society of Anaesthetists (NVA) [7,8]. POP during admission was managed with acetaminophen, naproxen, and in cases of expected severe pain with oxycodone. PONV prophylaxis for patients receiving general anaesthesia was intraoperatively managed with dexamethasone and granisetron. In case of PONV, available treatment options were granisetron, droperidol, metoclopramide, and domperidone.

Patients were discharged with written and verbal instructions by the nurse on how to manage POP and PONV and how to take care of the surgical wound. They received a medication box containing analgesic and antiemetic medication for three days postoperative. Two medication boxes were used in this study. In the case of minor surgical procedures, the patient received paracetamol, naproxen, pantoprazole, and metoclopramide. In the case of intermediate surgical procedures, long-acting oxycodone was added for the first postoperative day.

Patients were able to report their pain and nausea for up to seven days postoperative using the smartphone application 'OLVG Pain app'. The app sent three daily reminder notifications; however, the reporting was voluntary. Pain and nausea scores were displayed on the patients' electronic charts in the app. Data were obtained from the patients' medical records (Epic Systems Corporation. 2020. Epic Hyperspace). Data of patients active with the app were aggregated and displayed on a healthcare professional monitoring dashboard, which was observed daily by a medical assistant or physician assistant trained in pain and nausea assessments. If patients reported unbearable pain or nausea and requested assistance, they were contacted by message (via EPIC patient e-mail) or by telephone.

### Smartphone application

The OLVG pain app was developed in collaboration between patients, the patient council (advisory board of patients affiliated with the hospital), healthcare professionals, the Dutch Society of Anaesthesiology and Logicapps. As was commissioned by the Department of Anaesthesiology of OLVG Hospital. The usability of the application was evaluated in a proof of

concept study amongst 50 patients and 12 stakeholders (e.g. anaesthesiologists, patients from patient council) [5]. Both patients and stakeholders agreed that the application was easy to use, and its simplicity and design were well suited for pain recording. Furthermore, patients were willing and motivated to use the application for recording their pain. The difference in median pain intensity scores between those recorded by patients using the app and those recorded by nurses was not statistically significant. Two important recommendations from the proof of concept study have been implemented in the current version; the ability to also capture nausea as a patient-reported outcome and the direct transfer of the entered data to the electronic patient record. The app was designed for use on smartphones and tablets with Android 5.0 or higher and IOS 11.0 or higher operating systems. The following 'in-app' questions regarding postoperative pain were asked: Are you in pain? (Fig 1); How much pain do you have: 0 to 10 on numerical rating scale (NRS)? (Fig 2), Is your pain bearable? (Fig 3), Are you hindered by pain?, Do you feel something must be done to relieve your pain?. Nausea assessment in the app was based on the Myles Nausea Impact Scale [9], which assesses the presence of nausea, vomiting, and whether the patient requested treatment for nausea. The app automatically ended the 7-day postoperative follow-up with four closeout questions addressing overall pain, nausea, satisfaction with the app, and whether the medical assistant or physician assistant responded in time. The application was connected to the electronic medical record through a secure FHIR HL7 server connection.

## Outcome

The primary outcome was the incidence of patients with self-reported postoperative pain and nausea based on their response to the questions; Are you in pain?, Are you nauseous?, with the possible answers 'yes' or 'no' after daycare surgery at postoperative day (POD) 1 to 7.

Secondary outcomes for POD 1 up to POD 7 were: Pain intensity reported with the 'in-app' numeric rating scale. Incidence of unbearable postoperative pain and patients requiring additional pain management. Incidence of nausea and patients requiring additional treatment. Uptake of the application and differences in characteristics between active users and non-users of the application. Overall experienced pain, nausea and satisfaction with the application and received assistance.

## Data collection, Statistical analysis and reporting

Data were extracted from the OLVG data warehouse after running a query with the inclusion criteria with help of a software specialist (JR). Researchers BT, CS, and JB checked the data for completeness and manually added the missing patients or values. Additionally, we collected data on the socioeconomic status (SES) of the participants provided by Statistics Netherlands (CBS). The SES score is a representation of how municipalities, neighbourhoods, and communities in the Netherlands compare with each other. The average SES score is approximately 0; a higher score indicates that residents are more prosperous, have higher levels of education, and/ or are employed for longer periods. In this study, the score was used as an explanatory variable to examine differences between participants in the use of healthcare facilities such as the OLVG pain app. Data were analyzed using SPSS statistics (version 22.0 (IBM Corp. Released 2013. IBM SPSS Statistics for Windows, Version 22.0. Armonk, NY: IBM Corp.). Normally distributed data are shown as means and standard deviations (SD). Non-normally distributed data are shown as medians with range. Inter-group differences were tested using appropriate parametric and non-parametric tests. STrengthening the Reporting of OBservational studies in Epidemiology (STROBE) and REporting of studies Conducted using Observational Routinely-collected Data (RECORD) guidelines were followed to present the results in this paper [10,11].

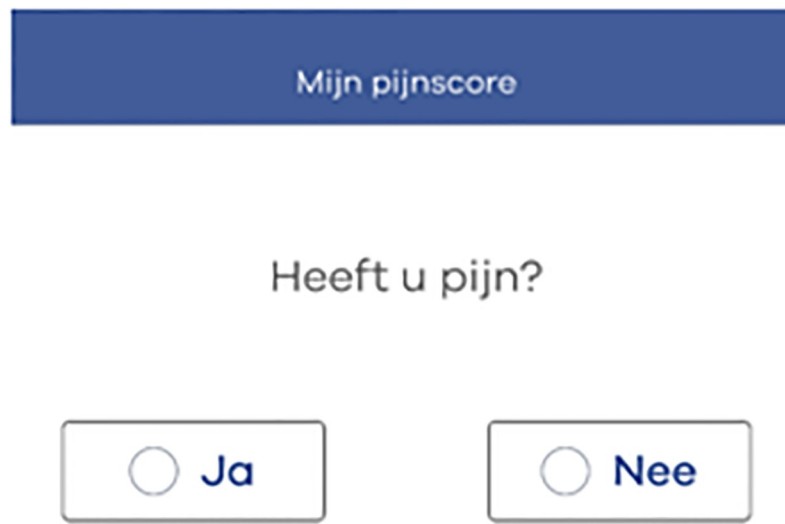

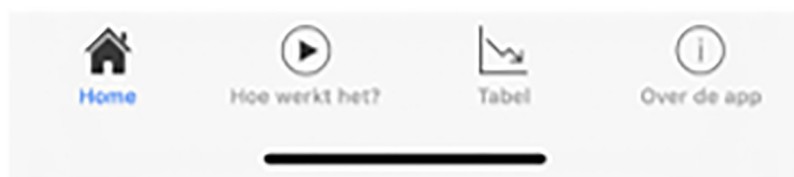

'My pain score' Are you in pain? (yes or no).

**Fig 1. Are you in pain?**

## Ethics

Ethical approval for this study (WO 20.239) was provided by the medical ethics committee (ACWO) and institutional board of directors of OLVG hospital, Amsterdam, the Netherlands (Chair of the boards Prof. dr. M.A.A.J. van den Bosch) on 14 January 2021. Furthermore, the

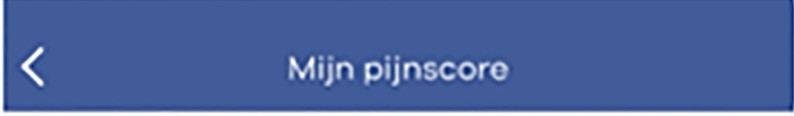

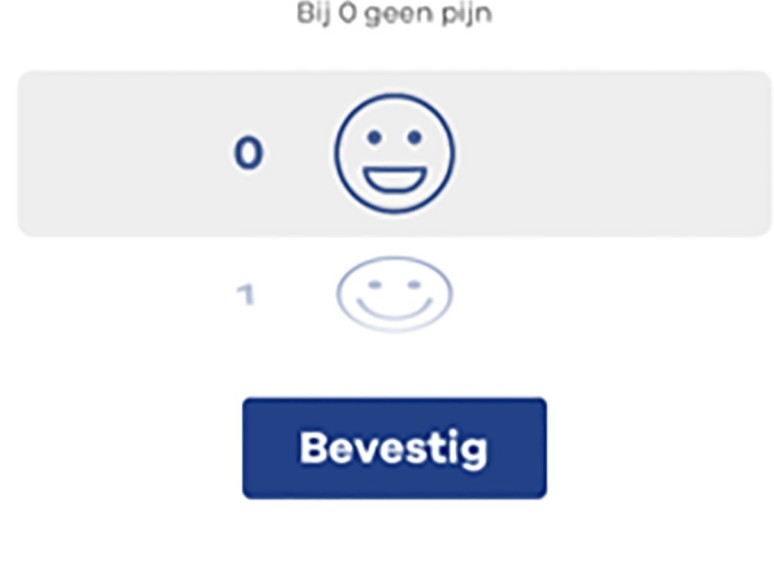

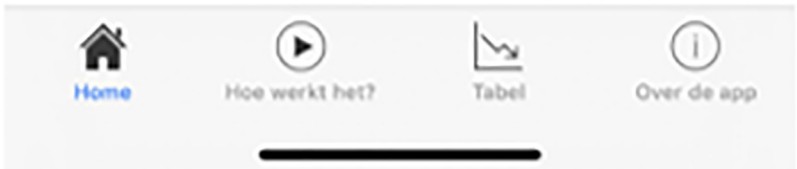

'My pain score' How much pain do you have? (NRS pain intensity wheel from 0; no pain to 10 worst imaginable pain).

**Fig 2. How much pain do you have: 0 to 10 on numerical rating scale (NRS)?**

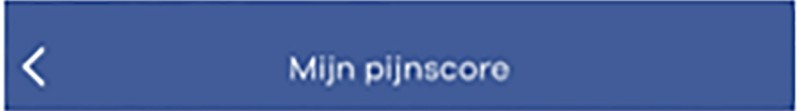

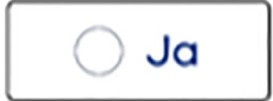 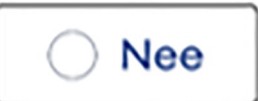

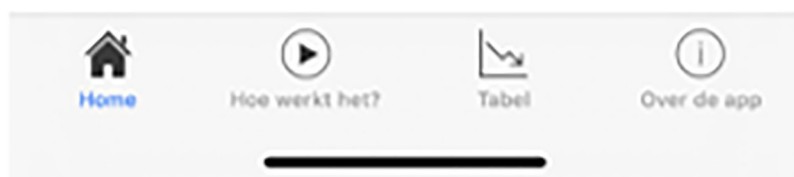

'My pain score' Is the pain bearable? (yes or no).

**Fig 3. Is your pain bearable?**

study was conducted according to the principles of Good Clinical Practice and in accordance with the declaration of Helsinki [12,13].

## Results

A total number of 4952 patients were included in the study. A total of 592 (12%) patients downloaded the application, of which 351 (7%) were unique active users, Patient flowchart

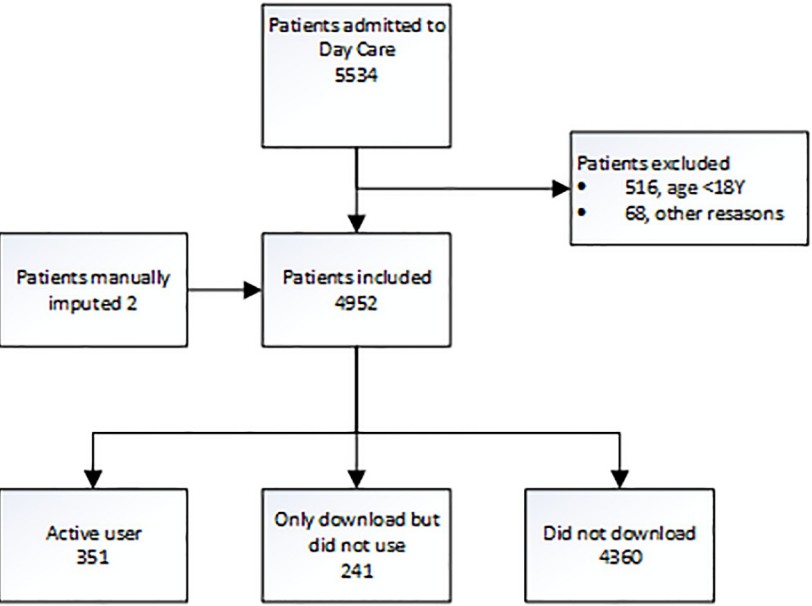

**Fig 4. Patient flowchart.**

(Fig 4). A total of 4360 (88%) patients refrained from downloading the application. Patient characteristics are presented in, Baseline characteristics of eligible patients admitted to day care (Table 1). A little more than half of patients were female 56% (2769), and the median age was 46 (18–92). Most patients had an American Society of Anesthesiologists Physical Status (ASA) of 1, 45% (2217) and 2, 42% (2069). The mean socio-economic-status (SES) score was -0.05434 (SD = 0.22).

**Table 1. Baseline characteristics of eligible patients admitted to day care.**

|  | Total cohort | Active | Not active | Not downloaded | P* |
|---|---|---|---|---|---|
| Cases, n (%) | 4952 | 351 (7%) | 241 (5%) | 4360 (88%) |  |
| Age, median (range) | 46 (18–92) | 44 (19–85) | 44 (18–82) | 46 (18–92) | .129† |
| Sex female, n (%) | 2769 (56%) | 222 (63%) | 164 (68%) | 2383 (55%) | .000‡ |
| ASA 1, n (%) | 2217 (45%) | 163 (47%) | 111 (46%) | 1943 (45%) | .318l§ |
| ASA 2, n (%) | 2069 (42%) | 145 (41%) | 100 (42%) | 1824 (42%) |  |
| ASA 3, n (%) | 634 (13%) | 42 (12%) | 27 (11%) | 565 (13%) |  |
| ASA 4, n (%) | 9 (<1%) | 0 | 0 | 9 (<1%) |  |
| SES, mean (SD) | -.05434 (.22) | -.01356 (.21) | -.05534 (,23) | -.05755 (.22) | .001¶ |
| **Surgical risk classification, n (%)** |  |  |  |  | .001‡ |
| Minor | 3950 (80%) | 254 (73%) | 184 (76%) | 3513 (81%) |  |
| Intermediate | 1000 (20%) | 96 (27%) | 57 (24%) | 847 (19%) |  |

*p-value of < 0.05 is considered statistically significant,

†Kruskal Wallis,

‡ Chi squared,

§Chi squared for trend,

¶ANOVA. ASA = American Society of Anesthesiologist, SES = social economic status, n = number

**Table 2. Overview of pain and nausea reporting of patients actively using the app.**

| Total number of patients actively using the app = 351 | Day 1 | Day 2 | Day 3 | Day 4 | Day 5 | Day 6 | Day 7 |
|---|---|---|---|---|---|---|---|
| Patients actively using the app | 229 (65%) | 182 (52%) | 160 (46%) | 135 (38%) | 131 (37%) | 108 (31%) | 100 (28%) |
| Median times used (range) | 2 (1 to 4) | 2 (1 to 8) | 2 (1 to 5) | 2 (1 to 5) | 2 (1 to 4) | 2 (1 to 7) | 2 (1 to 5) |
| Are you in pain?, n (%) | 174 (76%) | 112 (62%) | 105 (65%) | 79 (56%) | 70 (53%) | 56 (52%) | 44 (44%) |
| Indicate how much pain (highest pain score) | | | | | | | |
| NRS 1 to 3, n (%) | 34 (19%) | 24 (21%) | 38 (36%) | 31 (39%) | 29 (41%) | 16 (29%) | 13 (29%) |
| NRS 4 to 7, n (%) | 115 (66%) | 79 (71%) | 65 (62%) | 47 (59%) | 40 57%) | 38 (68%) | 30 (68%) |
| NRS 8 to 10, n (%) | 25 (14%) | 9 (8%) | 2 (2%) | 1 (1%) | 1 (1%) | 2 (4%) | 1 (2%) |
| Not bearable?, n (%) | 30 (9%) | 14 (4%) | 5 (5%) | 5 (6%) | 1 (1%) | 3 (3%) | 5 (11%) |
| Request to relief the pain, n (%) | 27 (8%) | 12 (3%) | 4 (4%) | 3 (4%) | 1 (1%) | 2 (4%) | 4 (9%) |
| Are you nauseous? n, (%) | 63 (28%) | 48 (26%) | 31 (19%) | 21 (16%) | 20 (15%) | 18 (17%) | 12 (12%) |
| Request to relief nausea, n (%) | 6 (10%) | 4 (8%) | 2 (6%) | 0 | 1 (5%) | 2 (11%) | 0 |

The total number of unique users is 351; the composition of daily users varies because some patients are not active with the application on a daily basis. For the variables; Are you in pain?; Not bearable?; Request to relief the pain; Are you nauseous?; and Request to relief nausea: n (%) represents the number and percentage of patients answering the question with an affirmative response. NRS = Numerical Rating Scale, n = number

General and injury-related surgeries represented 51% (2541) of the patients, and 80% (3950) of the surgical interventions were classified as minor surgical risks. There were statistically significant differences in sex, socioeconomic status, and surgical risk between the active user, non-active user, and non-downloader groups.

## Primary outcome

Postoperative pain (are you in pain = yes) was experienced by 174 (76%) of 229 active users on POD 1, Overview of pain and nausea reporting of patients actively using the app (Table 2). This decreased to 44 (44%) of the 100 active users on POD7, Incidence of postoperative pain (Fig 5). Similarly, postoperative nausea and vomiting (nauseous = yes) was experienced by 63 (28%) of 229 active users on POD1. This decreased to 12 (12%) of 100 active users on POD7, Incidence of postoperative nausea (Fig 6).

## Secondary outcome

In total, 351 unique patients were active users during the 7 days postoperative follow period, with an overall median times used of 2 (range, 1–8). An overview of the surgical specialism and 15 most performed surgeries in our cohort is provided in S1 Table: Overview surgical specialism and most proceeded interventions. The highest pain intensity scores of the active users were reported on POD 1 and POD 2: 25 (14%) patients and 9 (8%) patients with a numerical rating scale (NRS) score of 8 to 10. The number of patients who rated their pain as not bearable and requested for relief, decreased over the first 5 postoperative days, with 27 to 1 active users, respectively, (Fig 6). On POD 6 and 7, the number of patients with not bearable pain and a request for relief slightly increased to 5 and 4 patients, respectively. The values regarding the request for relief decreased from 6 (10%) patients on POD1 to 0 patients on POD7. It is important to emphasize that these outcomes solely represent the patient's responses to the in-app questions and do not confirm whether they actually received any additional attention or care.

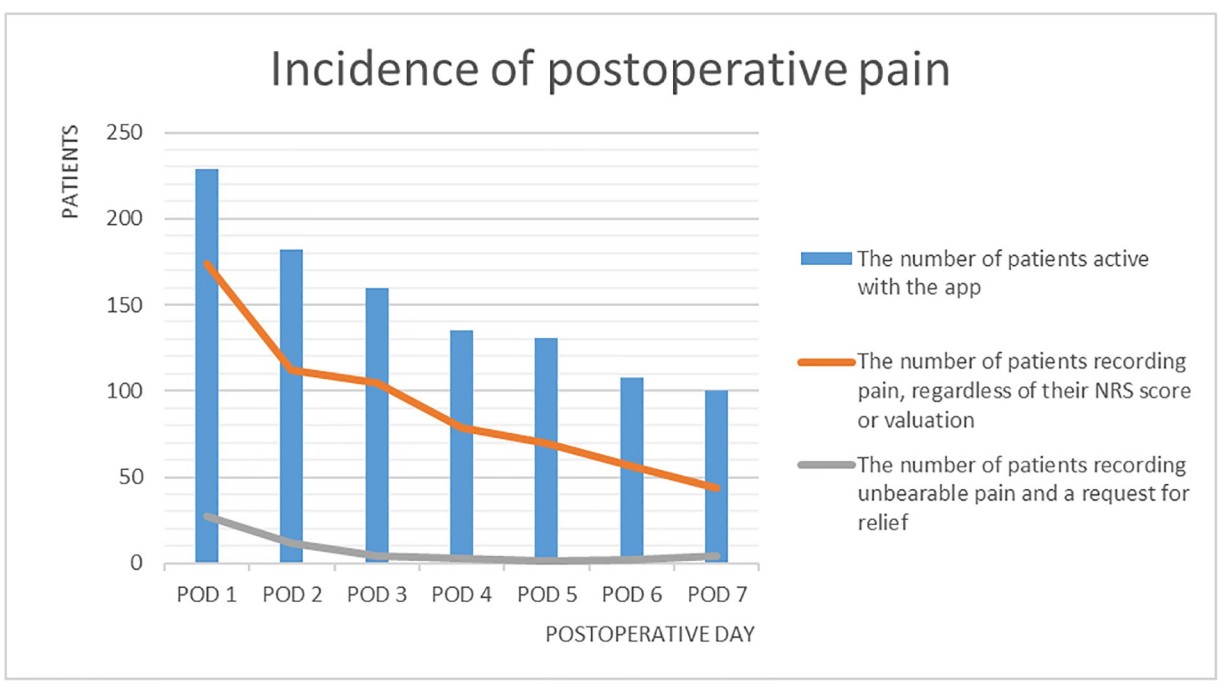

**Fig 5. Incidence of postoperative pain.**

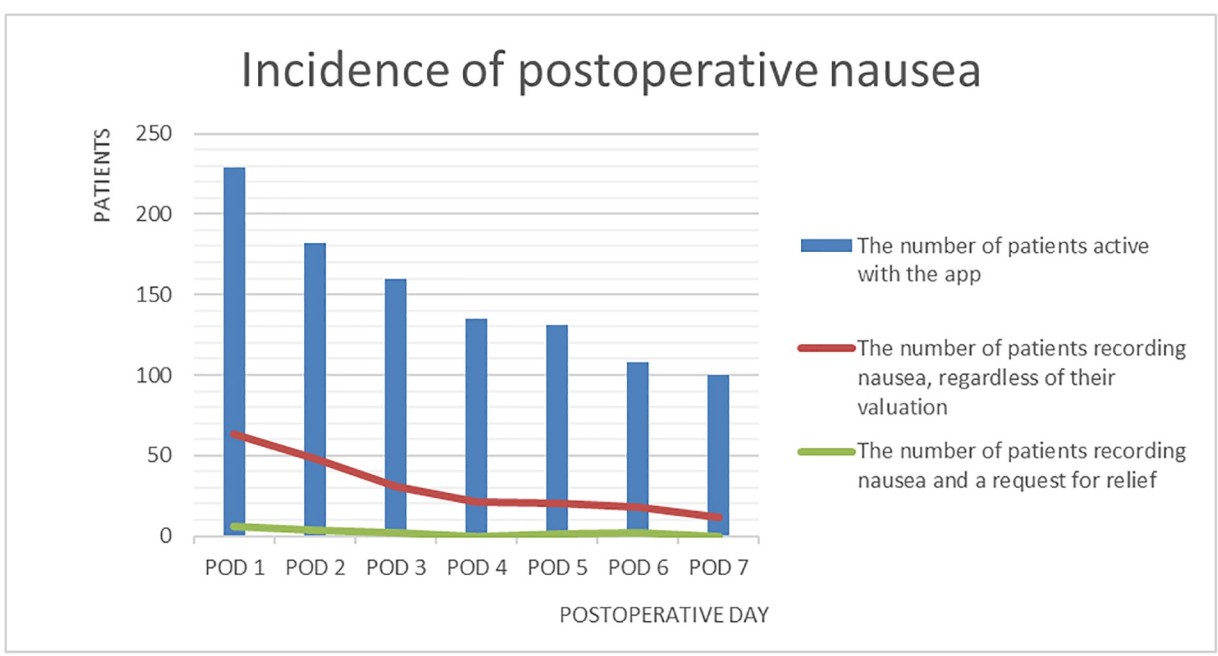

**Fig 6. Incidence of postoperative nausea.**

## Overall experienced pain, nausea, satisfaction with the application and received assistance

The in-app closeout questions on POD 7 were completed by 60 (17%) active users, valuing experienced pain, nausea, the remote monitoring tool, and received assistance. Regarding experienced pain, on a 5 point Likert scale, 55 (92%) patients reported no pain at all, mild pain, or bearable pain. Reflecting on the initial 7 days post-surgery, none of the patients who completed the in-app closeout questions indicated the presence of severe pain. Regarding nausea 54 (90%) patients reported no, little, or bearable nausea. One of the patients experienced severe nausea. Most patients were satisfied with the app 57 (95%) and reported that it was okay, pleasant, or very pleasant to record their pain and nausea with it, only a minority of 3 (5%) patients reported that it was not pleasant or unpleasant. The received assistance was rated as very pleasant, pleasant, or Ok by 56 (93%) patients, while 4 (7%) patients rated it as not pleasant or very unpleasant.

## Discussion

In this retrospective cohort study, we evaluated remote monitoring of pain and PONV after daycare surgery using a dedicated smartphone application (the OLVG Pain app). This evaluation was performed one year after the implementation of the app in daily practice. Our main finding was that only 12% of eligible patients downloaded the application, and that no more than 7% of all patients actually used the application to report pain and nausea. Among the active users, most pain and nausea were reported in the first two days after discharge.

Our finding that the proportion of patients recording pain was highest on POD 1 and 2, with up to 14% of recordings indicating severe pain, is in line with the available literature. However, previous studies showed an overall higher prevalence of moderate-to-severe pain of 30% on POD1 [14,15] and up to 43% on POD 7 [16]. This difference may be due to the chosen threshold for moderate to severe pain and the patient study population.

Regarding PONV, we found a relatively high proportion of patients who experienced nausea; however, only a small percentage requested relief. Few studies have evaluated PONV after daycare surgery, demonstrating similar incidences, varying from 28% to 57% [17–20]. The heterogeneity of the methodologies and our low response rate make it difficult to compare the results. Only one study reported PONV rates up to POD7, a prospective observational cohort study that included 239 patients and found that 6% of the patients experienced nausea on POD7 [18].

Despite the promising results of our previous proof-of-concept study [5], the uptake and use of the application was surprisingly low. This is contrast with the results of the SATELIA application [21] which is the most comparable app with direct electronic record integration, and pain and side effect monitoring. Notably, it achieved a high rate of active users on POD1, potentially due to proactive nurse follow-up. We recognize this from our own results that active motivation of patients by one of the researchers present in the day care ward during the last 3 months of the follow-up period has resulted in 285 out of 351 patients actively engaging with the application.

We anticipated that more patients would have used the application after discharge. We had identified and addressed important implementation themes from a meta viewpoint namely; size of the hospital, top management support, organizational readiness, centralization in decision-making and absorptive capacity [22,23]. We found that the most important factor influencing the adoption of the OLVG pain app was absorptive capacity, primarily determined by the innovation's urgency, relevance, and a sense of ownership and responsibility. We addressed this by; enhancing the innovation's relevance, involving stakeholders from the

outset, and assigning ambassadors and managers to support stakeholder engagement and to provide proper guidance and training [24].

Possibly we are still facing, with regards to Rogers's diffusion of innovation theory (DOI), that patients who are actively using the application are the 'innovators' and 'early adopters' and that we still have not passed the point of critical mass for innovation adoption amongst our patients [25].

One potential challenge that may have been overlooked is the digital divide, which is linked to pre-existing social inequalities as a contributing factor to non-adherence to e-health [26]. This accounts for OLVG Hospital patients, as these locations are situated in low socio-economic neighborhoods of Amsterdam [27]. The mean socioeconomic score (SES) of the patients in our study was below 0, indicating that they were less prosperous, had lower levels of education, and were employed for shorter periods than the average resident in the Netherlands. Therefore, it is plausible that many of our patients lacked access to or were unable to use the remote-monitoring app.

This is supported by the results of a recent study conducted in Amsterdam among mothers from low-SES backgrounds, which shows that poverty and the complexity of information and communication technology, influence perspectives, experiences, and ICT-related needs [28].

This present study has some limitations, including those inherent to observational cohort studies that could affect our data and outcomes [29]. Furthermore, the COVID-19 pandemic was still ongoing during the study period, which is likely to have impacted our study results. Measures such as social distancing and the deployment of surgical and anesthesia staff in covid care have resulted in cancellations or postponement of daycare surgeries. In relation to this, the presence of a researcher on the day care ward, who could offer additional guidance to the patients about the application and provide instructions on its usage, was only allowed during the last 3 months of the study's observation period. This resulted in fewer patients in the study cohort. Those who were admitted underwent more urgent surgical procedures, which explains the high percentage of trauma patients in our data.

In brief, to compare of the OLVG pain app with other recently published results of applications designed for post-operative follow-up [21,30–32]. The Mserv application [30] is an Android-exclusive app for thoracic and urogenital procedures, active within hospital setting. In contrast, the OLVG pain app that is accessible to outpatients and clinically admitted patients and is compatible with both iOS and Android operating systems. RecoverWell application [31], which was studied in a breast cancer surgery trial, offers advantages such as photo follow-up and drain monitoring but lacks the crucial feature of integration with electronic medical records, a feature that the OLVG pain app has. The Q1.6 Inguinal Hernia app [32] uses 'twitch crowdsourcing' for real-time data collection but does not provide patient follow-up. In addition to these, there are other recent applications that focus more on rehabilitation and revalidation for specific surgical procedures, featuring built-in patient instructions and reminders rather than primarily focusing on the management of postoperative pain and nausea [33–35].

One of the advantages of our application is that the outcomes of pain and nausea were self-recorded by the patient at home without interference from a healthcare professional. It is known that both the patient and healthcare professionals alter the pain and nausea recording for their benefit or adjust them to subjective perceptions during interactive assessments [36,37]. Therefore, we still believe that self-recording is likely to provide less biased and more realistic outcome data. However, the low number of active users and irregular recording may limit the interpretation of the results. Future research should focus on the uptake of mobile health and remote monitoring, and whether it benefits the quality of recovery.

## Conclusion

Most patients active with the application experienced pain and nausea on the first and second days after discharge, but only a small minority of eligible patients used the application. Those who used it were satisfied with the possibilities offered to them. Future research should focus on increasing the uptake and effect of this application on the quality of recovery.

## Supporting information

**S1 Table. Overview surgical specialism and most proceeded interventions.**
(DOCX)

**S1 STROBE Checklist. STROBE Statement: Monitoring of Self-Recorded Pain and Nausea via Smartphone Following Day Care Surgery, First Year Results: An Observational Cohort Study.**
(DOC)

## Author Contributions

**Conceptualization:** Bram Thiel, Cor Kalkman, Marc Godfried.

**Data curation:** Bram Thiel, Chiem Seesing.

**Formal analysis:** Bram Thiel, Jamey Blaauboer.

**Funding acquisition:** Bram Thiel.

**Investigation:** Bram Thiel, Jamey Blaauboer, Chiem Seesing.

**Methodology:** Bram Thiel.

**Project administration:** Bram Thiel.

**Software:** Jamshid Radmanesh.

**Supervision:** Cor Kalkman, Marc Godfried.

**Validation:** Bram Thiel, Jamey Blaauboer, Chiem Seesing.

**Writing – original draft:** Bram Thiel, Seppe Koopman, Marc Godfried.

**Writing – review & editing:** Seppe Koopman, Cor Kalkman, Marc Godfried.

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
