## [Decision Letter · Decision Letter 0]

8 Sep 2023

PDIG-D-23-00297

Patient self-reported pain and nausea via smartphone following daycare surgery, first year results: An observational cohort study

PLOS Digital Health

Dear Dr. Thiel,

Thank you for submitting your manuscript to PLOS Digital Health. After careful consideration, we feel that it has merit but does not fully meet PLOS Digital Health's publication criteria as it currently stands. Therefore, we invite you to submit a revised version of the manuscript that addresses the points raised during the review process.

Please submit your revised manuscript within 60 days Nov 07 2023 11:59PM. If you will need more time than this to complete your revisions, please reply to this message or contact the journal office at digitalhealth@plos.org. Please include the following items when submitting your revised manuscript:

We look forward to receiving your revised manuscript.

Kind regards,

Haleh Ayatollahi

Section Editor

PLOS Digital Health

Journal Requirements:

1. In the Funding Information you indicated that no funding was received. Please revise the Funding Information field to reflect funding received.

2. Please provide separate figure files in .tif or .eps format only and remove any figures embedded in your manuscript file. Please also ensure that all files are under our size limit of 10MB.

3. We do not publish any copyright or trademark symbols that usually accompany proprietary names, eg ©, ®, ™ (e.g. next to drug or reagent names). Please remove all instances of trademark/copyright symbols throughout the text, including ® on page 5.

Additional Editor Comments (if provided):

Reviewers' comments:

Reviewer's Responses to Questions

**Comments to the Author**

1. Does this manuscript meet PLOS Digital Health’s publication criteria? Is the manuscript technically sound, and do the data support the conclusions? The manuscript must describe methodologically and ethically rigorous research with conclusions that are appropriately drawn based on the data presented.

Reviewer #1: Yes

Reviewer #2: Yes

Reviewer #3: Partly

Reviewer #4: No

2. Has the statistical analysis been performed appropriately and rigorously?

Reviewer #1: Yes

Reviewer #2: Yes

Reviewer #3: Yes

Reviewer #4: No

3. Have the authors made all data underlying the findings in their manuscript fully available (please refer to the Data Availability Statement at the start of the manuscript PDF file)?

Reviewer #1: Yes

Reviewer #2: Yes

Reviewer #3: No

Reviewer #4: Yes

4. Is the manuscript presented in an intelligible fashion and written in standard English?

Reviewer #1: Yes

Reviewer #2: Yes

Reviewer #3: Yes

Reviewer #4: Yes

5. Review Comments to the Author

Reviewer #1: The manuscript of Thiel et al describes a retrospective study of patients self-reported pain and nausea after daycare surgery using a smartphone application. Digital solutions have the potential to improve the interactions between medical professionals and their patients. The present study evaluated the adoption rate and post-operative complications (pain, nausea) of patients by self-reported ratings. Overall, the results are of great interest for the growing community of digital health professionals. The manuscript could be improved by clarifying the issues stated below. 

Major issues

1) This is not the first app that was developed to monitor and/or report post-operative complications. Please include a paragraph where you compare your app to other common alternatives. 

1a) State the advantages (e.g. language, data integration, unique features) and disadvantages of your software when compared to the existing alternatives.

1b) The frequency of patients that were in pain or feeling nauseous and nothing was done is quite high in this study (see Figure 2 and Figure 3). In the beginning it is stated that the use of an smartphone application could be advantageous for improving patient care. 

2) Please discuss possible reasons for this observation. Are changes to the implementation necessary and if so which ones would be most beneficial for the patient?

Consider including an analysis of patient satisfaction in the subgroups where something was done and nothing was done.

Minor issues

line 98: Smartphone requirements, ability to understand the Dutch language and cognitive impairment (e.g. dementia) are likely to lead to lower adoption to the proposed digital application. However, those patients were not excluded in the present study. The limitations should state that these factors were not assessed and could lead to a reduction in the adoption rate.

line 102: Was there a specific reason that the researcher was only present for the last 3 months of the observation period? And did the presence of the researcher lead to time-dependent differences of active users during this period?

Table 1: The subtitle of the table should be re-written. I would recommend that you use superscript symbols (e.g.†,¶,…) behind p-values in the table. In the the annotation of the table you can write the corresponding test for each symbol. The current style is confusing for the reader.

Furthermore, in the methods section it is written that median and IQR are used for non-normal distributed variables. In the table median and range is used. Please decide what strategy you want to use.

In the SES row the commata should probably be substitutes with dots

line 223: Consider changing the statement „None of the patients experienced any severe pain.“ to „Retrospectively, none of the patients reported that they experienced any severe pain.“. In line 213 it is mentioned that 14% reported NRS scores above 7, so either all those patients dropped out or more likely they did not remember their pain as vividly. 

line 233: The 11% is in contrast to the 12% mentioned earlier.

Reviewer #2: The conclusion of the study presented in this retrospective cohort investigation reflects a comprehensive evaluation of remote monitoring through a specialized smartphone application (the OLVG Pain app) in the context of pain and postoperative nausea and vomiting (PONV) management after outpatient surgery. The analysis took place one year following the app's integration into routine clinical practice. The primary findings unveiled a rather modest adoption rate, as only 11% of eligible patients actually downloaded the app, and merely 7% of the total patient pool engaged with it for pain and nausea reporting. Among active users, most symptom reports occurred within the initial two days post-discharge.

The study's identification of the highest proportion of pain reports on postoperative days (POD) 1 and 2, with up to 14% indicating severe pain, aligns with existing literature. Notably, the prevalence of moderate-to-severe pain reported in this study, at 14% on POD 1, differs from earlier research showing higher rates, potentially due to variances in pain assessment thresholds and the patient demographics under examination.

Regarding PONV, the research identifies a relatively elevated occurrence of nausea among patients, yet a small percentage sought relief. Comparable studies in the realm of PONV after ambulatory surgery have demonstrated similar incidences, though methodological disparities and the study's relatively low response rate hinder direct comparison. The sole study assessing PONV up to POD 7 exhibited a 6% nausea occurrence rate on that day, signifying the need for more comprehensive investigations.

Despite the encouraging outcomes of a prior proof-of-concept study, the actual implementation and utilization of the application proved surprisingly subdued. Anticipations of greater post-discharge usage were not met, prompting consideration of Rogers's diffusion of innovation theory, which suggests that current users are early adopters and innovators, and broader adoption is yet to be achieved. A potentially underestimated factor, the digital divide, comes to the forefront as a potential constraint, particularly given the socioeconomic background of OLVG Hospital patients, primarily residing in economically disadvantaged Amsterdam neighborhoods. This raises concerns about equitable access and capability to utilize e-health solutions, particularly among the less privileged.

The research acknowledges its limitations, inherent to observational cohort studies, and highlights the ongoing impact of the COVID-19 pandemic on study outcomes, given its influence on surgical scheduling and patient demographics. The advantages of self-reported outcomes, untainted by healthcare professional influence, are noted. However, the limited user engagement and sporadic symptom reporting impose constraints on result interpretation. Future inquiries should delve into the acceptance and effectiveness of mobile health applications for remote monitoring in the context of recovery.

In conclusion, the study offers valuable insights into the implementation and outcomes of remote monitoring via a specialized smartphone application for pain and PONV management post-outpatient surgery. The investigation underscores the challenges of adoption and the potential influence of socioeconomic factors, in addition to recognizing the impact of ongoing global events. These findings emphasize the significance of continued research to elucidate the broader implications and benefits of mobile health solutions in optimizing patient care.

Reviewer #3: The paper aims to show the results of a one year study of post-operative patients using a pain reporting app. 

1) I cannot access the supplemental files -- 

2) As a matter of style, please start sentences with letters and avoid starting with numbers (eg, "56% of patients...")

3) Line 263: there is mention of first generation and non-Western types of patients but there is no direct study data to support this - these were linked to the SES demographic data (summarized) rather than to the specific individual data. Suggest to remove these observations. 

4) Were patient feedback on the usability of the app obtained? If yes, that will be an important statement. If patient feedback were not, then that could be stated briefly as well.

5) Line 129: mismatch between "a patient" and "they" -- pls match the plurality

6) Patients were given medication (minor) to take home. Did the app document if the patient took them regularly? Or just that a patient intervention (generic) was done?

7) Of the patients who were on oxycodone, were these directly observed by a health professional prior to discharge or were these self-administered? Pls mention as this could have an impact on the pain scores.

The paper can contribute significantly on the knowledge about mobile app uptake of patients. Suggest referencing either the UTAUT and Delone-Mclean frameworks to identify which aspects of technology adoption were covered by the study and which were not.

Reviewer #4: Thank you for the opportunity to review this manuscript entitled “Patient self-reported pain and nausea via smartphone following daycare surgery, first year results: An observational cohort study”. 

There has certainly been a shift towards elective daycase surgery which provides an attractive cohort of patients that would potentially benefit from digital health interventions.

Unfortunately, however I do not believe the article provides sufficient methodological or clinical benefit that I would be able to recommend it for publication in PLOS Digital Health, even in the case of substantial revision. Further justification for this decision is outlined below:

The included primary outcome was the incidence of patients with self-reported pain and nausea in the early post-operative period – however this doesn’t really provide much information about the clinical utility of the app, i.e. ultimately we want to how the app influences treatment compared to standard practice. This information is present in very small numbers in terms of those that have pain/nausea and have had something done, but there is no way to compare with traditional pathways, and the numbers included are insufficient to draw any meaningful conclusions.

There was limited information included as to the types of clinical cases performed, which would likely have a significant impact on the perceived rates of pain and nausea. For example, general surgical patients would likely have significant differences in the subsequent levels of nausea compared to orthopaedic patients. Heterogeneity of this population makes it very difficult to draw any useful information regarding rates for more specific sub-populations where clinical teams are likely to be working.

There was a quite dramatic lack of uptake for the number of patients included in the study. This again limits the potential generalisability of study findings given such a small sub-sample responded, and that there is likely to be significant bias within this population of responders. It would likely be far more useful to an audience to try in much greater depth to identify why the uptake was so poor. This would likely provide greater reach of utility, for example providing potential key information to those current going through the process of app development or similar attempts at clinical practice integration.

6. PLOS authors have the option to publish the peer review history of their article (what does this mean?). If pu

---

## [Decision Letter · Decision Letter 1]

13 Dec 2023

PDIG-D-23-00297R1

Patient self-reported pain and nausea via smartphone following day care surgery, first year results: An observational cohort study

PLOS Digital Health

Dear Dr. Thiel,

Thank you for submitting your manuscript to PLOS Digital Health. After careful consideration, we feel that it has merit but does not fully meet PLOS Digital Health's publication criteria as it currently stands. Therefore, we invite you to submit a revised version of the manuscript that addresses the points raised during the review process.

Please submit your revised manuscript within 30 days Jan 12 2024 11:59PM. If you will need more time than this to complete your revisions, please reply to this message or contact the journal office at digitalhealth@plos.org. Please include the following items when submitting your revised manuscript:

We look forward to receiving your revised manuscript.

Kind regards,

Haleh Ayatollahi

Section Editor

PLOS Digital Health

Journal Requirements:

Additional Editor Comments (if provided):

I appreciate the authors for thier time and efforts to revise the manuscript. Please address the following comments in your next revision, too.

1- Please add appropriate keywords after the abstract.

2- In the introduction section, please review other similar applications that have been developed so far.

3- Some figures of the application should be added to the manuscript.

Reviewers' comments:

Reviewer's Responses to Questions

**Comments to the Author**

1. If the authors have adequately addressed your comments raised in a previous round of review and you feel that this manuscript is now acceptable for publication, you may indicate that here to bypass the “Comments to the Author” section, enter your conflict of interest statement in the “Confidential to Editor” section, and submit your "Accept" recommendation.

Reviewer #3: All comments have been addressed

Reviewer #5: All comments have been addressed

2. Does this manuscript meet PLOS Digital Health’s publication criteria? Is the manuscript technically sound, and do the data support the conclusions? The manuscript must describe methodologically and ethically rigorous research with conclusions that are appropriately drawn based on the data presented.

Reviewer #3: Yes

Reviewer #5: Partly

3. Has the statistical analysis been performed appropriately and rigorously?

Reviewer #3: Yes

Reviewer #5: I don't know

4. Have the authors made all data underlying the findings in their manuscript fully available (please refer to the Data Availability Statement at the start of the manuscript PDF file)?

Reviewer #3: Yes

Reviewer #5: Yes

5. Is the manuscript presented in an intelligible fashion and written in standard English?

Reviewer #3: Yes

Reviewer #5: Yes

6. Review Comments to the Author

Reviewer #3: (No Response)

Reviewer #5: -----

7. PLOS authors have the option to publish the peer review history of their article (what does this mean?). If published, this will include your full peer review and any attached files.

**Do you want your identity to be public for this peer review?** For information about this choice, including consent withdrawal, please see our Privacy Policy. 

Reviewer #3: No

Reviewer #5: No

---

## [Decision Letter · Decision Letter 2]

1 Feb 2024

PDIG-D-23-00297R2

Patient self-reported pain and nausea via smartphone following day care surgery, first year results: An observational cohort study

PLOS Digital Health

Dear Dr. Thiel,

Thank you for submitting your manuscript to PLOS Digital Health. After careful consideration, we feel that it has merit but does not fully meet PLOS Digital Health's publication criteria as it currently stands. Therefore, we invite you to submit a revised version of the manuscript that addresses the points raised during the review process.

Please submit your revised manuscript within 30 days Mar 02 2024 11:59PM. If you will need more time than this to complete your revisions, please reply to this message or contact the journal office at digitalhealth@plos.org. Please include the following items when submitting your revised manuscript:

We look forward to receiving your revised manuscript.

Kind regards,

Haleh Ayatollahi

Section Editor

PLOS Digital Health

Journal Requirements:

Additional Editor Comments (if provided):

Reviewers' comments:

Reviewer's Responses to Questions

**Comments to the Author**

1. If the authors have adequately addressed your comments raised in a previous round of review and you feel that this manuscript is now acceptable for publication, you may indicate that here to bypass the “Comments to the Author” section, enter your conflict of interest statement in the “Confidential to Editor” section, and submit your "Accept" recommendation.

Reviewer #6: All comments have been addressed

2. Does this manuscript meet PLOS Digital Health’s publication criteria? Is the manuscript technically sound, and do the data support the conclusions? The manuscript must describe methodologically and ethically rigorous research with conclusions that are appropriately drawn based on the data presented.

Reviewer #6: Yes

3. Has the statistical analysis been performed appropriately and rigorously?

Reviewer #6: Yes

4. Have the authors made all data underlying the findings in their manuscript fully available (please refer to the Data Availability Statement at the start of the manuscript PDF file)?

Reviewer #6: No

5. Is the manuscript presented in an intelligible fashion and written in standard English?

Reviewer #6: Yes

6. Review Comments to the Author

Reviewer #6: This paper provides a description of a smartphone app allowing day surgery patients to self-report post-operative pain and nausea, and receive health professional intervention for distressing levels of symptoms. The authors found a relatively low uptake (7% of the study population became active users of the app), but among those found that pain and nausea were most common on post-op days 1 and 2, and fell off quickly afterwards. This is an important contribution to the growing field of smartphone apps for healthcare interaction. The utility of the app is demonstrated, and the paper lays groundwork for future studies regarding the role of patient preferences in leveraging digital tools for streamlined access to the healthcare system, and likely provides a baseline for future interventional studies aimed at modifying pain and nausea symptoms after discharge. 

I have several additional comments:

1. Raw data is not shared, citing GDPR. Although there are many institutional and legal barriers to sharing healthcare data, data sharing is critical to promote reproducible research. Could the authors share a de-identified repository of their raw data?

2. The authors suggest low socioeconomic status contributed to low uptake of the smartphone app. This is certainly possible. However, in the current digital climate many consumers have concerns about data privacy and the value-added of downloading additional apps to their phone. Do the authors feel that these “savvy consumer” concerns may have impacted uptake as well? What was the data privacy policy associated with the app (e.g., was the data transmitted or maintained via any commercial 3rd party?), and were prospective patients counseled on this? Should consideration be given to alternate digital formats, e.g. a standard website vs an app, or text message communication to collect the same data?

3. Please explain what is meant by “the patient council” in line 135

4. Line 166-167 – should this say “with the help of a software specialist”?

5. Line 195 – suggest “injury-related” rather than “traumatic” surgeries 

6. Table 2 – for the bottom four rows, please clarify (in caption or table) if values represent the number of people who answered the question at all on a particular day, or the number with an affirmative response. 

7. Line 218 – is the median times used overall or per day?

8. There are several areas of the paper that require copy editing for standard English, in particular lines 69 and 70 (suggest present tense verbs), 96 (ordering of month/date is different for American and European audiences, suggest just writing out the month and date), 157 (primary outcome was the incidence), 160-164 (these are list items and are not full sentences, suggest separate with semicolon), 222 (“not bearable and requested relief”), 223 (non-bearable vs no bearable?), 230 (clarify), 233 (extra “.”), 264 (“would use” the app), 280 (“average resident”)

7. PLOS authors have the option to publish the peer review history of their article (what does this mean?). If published, this will include your full peer review and any attached files.

**Do you want your identity to be public for this peer review?** For information about this choice, including consent withdrawal, please see our Privacy Policy. 

Reviewer #6: No

---

## [Decision Letter · Decision Letter 3]

22 May 2024

Patient self-reported pain and nausea via smartphone following day care surgery, first year results: An observational cohort study

PDIG-D-23-00297R3

Dear Thiel,

We are pleased to inform you that your manuscript 'Patient self-reported pain and nausea via smartphone following day care surgery, first year results: An observational cohort study' has been provisionally accepted for publication in PLOS Digital Health.

Best regards,

Haleh Ayatollahi

Section Editor

PLOS Digital Health

Reviewer Comments (if any, and for reference):

Reviewer's Responses to Questions

**Comments to the Author**

1. If the authors have adequately addressed your comments raised in a previous round of review and you feel that this manuscript is now acceptable for publication, you may indicate that here to bypass the “Comments to the Author” section, enter your conflict of interest statement in the “Confidential to Editor” section, and submit your "Accept" recommendation.

Reviewer #6: All comments have been addressed

2. Does this manuscript meet PLOS Digital Health’s publication criteria? Is the manuscript technically sound, and do the data support the conclusions? The manuscript must describe methodologically and ethically rigorous research with conclusions that are appropriately drawn based on the data presented.

Reviewer #6: Yes

3. Has the statistical analysis been performed appropriately and rigorously?

Reviewer #6: Yes

4. Have the authors made all data underlying the findings in their manuscript fully available (please refer to the Data Availability Statement at the start of the manuscript PDF file)?

Reviewer #6: No

5. Is the manuscript presented in an intelligible fashion and written in standard English?

Reviewer #6: Yes

6. Review Comments to the Author

Reviewer #6: Thank you to the authors for their clarifications and additions to the manuscript.

7. PLOS authors have the option to publish the peer review history of their article (what does this mean?). If published, this will include your full peer review and any attached files.

**Do you want your identity to be public for this peer review?** For information about this choice, including consent withdrawal, please see our Privacy Policy.

Reviewer #6: No
